

# The prospective COVID-19 vaccine: willingness to pay and perception of community members in Ibadan, Nigeria

Olayinka Ilesanmi[1,2], Aanuoluwapo Afolabi[1] and Obioma Uchendu[1,2]

[1] Department of Community Medicine, University of Ibadan, Ibadan, Oyo, Nigeria
[2] Department of Community Medicine, University College Hospital, Ibadan, Oyo, Nigeria

Corresponding author
Aanuoluwapo Afolabi,
afoannade@gmail.com

## ABSTRACT

**Background:** The introduction of the COVID-19 vaccine necessitates the assessment of individual perception regarding the vaccine. This study aimed to assess the perception of community members and willingness to pay for the prospective COVID-19 vaccine in Ibadan, Nigeria.

**Methods:** A descriptive cross-sectional study design was used. Data were collected using an interviewer-administered questionnaire in September 2020. We studied community members aged 15 years and above using a multi-stage sampling technique. The perceptions of respondents about the COVID-19 vaccine were assessed on eight questions using the five-point Likert scale with a score point of "1" assigned for "Strongly Agree", "2" for "Agree", "3" for "Not decided", "4" for "Disagree", and "5" for "Strongly disagree". During analysis, we reverse-coded the options by assigning a point of "1" for "Strongly disagree", "2" for "Disagree", "3" for "Not decided", "4" for "Agree", and "5" for "Strongly disagree". However, questions asked in the negative directions were not reverse-coded during analysis. Eight questions were used to assess the perception of community members regarding the prospective COVID-19 vaccine, and overall, the maximum point was 40. Points greater than or equal to 32 points (80%) implied positive perception. Descriptive statistics were done. Chi-square tests were used for the assessment of associations between sociodemographic characteristics and willingness to pay for the prospective COVID-19 vaccine. We conducted logistic regression tests on statistically significant variables at $p$-values <0.05.

**Results:** The mean age of the 440 respondents studied was 37.22 ± 15.36 years, 193 (49.00%) were males, and 292 (67.30%) of the respondents had heard of the prospective COVID-19 vaccine. Among them, 232 (79.50%) respondents had positive perception regarding COVID-19 vaccine. Individuals in the fifth wealth quintile were ten times more likely to be willing to pay for the prospective COVID-19 vaccine compared to those in the first wealth quintile (Adjusted Odds Ratio = 9.57, 95% CI [2.88–31.82], $p$ = <0.01).

**Conclusion:** The prospective COVID-19 vaccine should be subsidized or made freely available to everyone.

## INTRODUCTION

The 2019-Coronavirus disease (COVID-19) is a droplet infection characterized by rapid transmission, high mortality rate, and resulting complications among humans globally (*Al-Hanawi et al., 2020*). Due to these features, COVID-19 was declared a global pandemic by the World Health Organization (WHO), and this necessitated the implementation of non-pharmaceutical control measures by all countries around the globe (*WHO, 2020*). These control measures have included the use of face masks, social distancing, school lockdowns, border closure, and hygiene protocols (*Ilesanmi, Oguntoye & Afolabi, 2020*; *Ilesanmi, Ariyo & Afolabi, 2020*; *NCDC, 2020*). In spite of the above-mentioned containment and control efforts, COVID-19 has remained a global threat with 100,383,705 cases and 2,152,357 deaths recorded as of 25th January, 2021 of which the African continent makes up 3.5% of cases and 4% mortality. Of this global total, Nigerian has reported 122,996 cases and 1,507 deaths as of the reference date (*Worldometer, 2020*). The daily rise in COVID-19-related cases and fatalities thus indicate the inadequacy of the present COVID-19 mitigation measures. This therefore reveals the need for the development of vaccines for the aversion of further spread of COVID-19 locally and globally, a task for which individual perception needs to be assessed.

Vaccines have demonstrated an excellent historical capacity for the elimination of many infectious illnesses such as tetanus, diphtheria, polio, rabies, pertussis, measles, and yellow fever (*Chukwuocha et al., 2018*). The routine immunization program and the expanded program on immunization have enabled the number of persons covered for immunization (*Chukwuocha et al., 2018*). These programs have represented great feats in the prevention of common childhood illnesses and the maintenance of the well-being of children. In the context of malarial infection, the development of an efficacious malarial vaccine has been suggested as a vital strategy for reducing the burden of malaria especially in malarial-endemic countries such as Nigeria and Ghana (*Ojakaa et al., 2011*). The RTS,S malaria vaccine has been developed, and is being researched to evaluate its efficacy (*Ojakaa et al., 2011*). The development of a safe and effective vaccine against the Ebolavirus disease (EVD) has been identified as an important tool for the prevention of future EVD outbreaks (*Ojakaa et al., 2011*; *Huo et al., 2016*). In lieu of this, experimental vaccines on EVD have commenced in five districts in Sierra Leone where majority of EVD cases have been recorded. Vaccine development however introduces new interventions which may be associated with some challenges (*Huo et al., 2016*).

Challenges have been experienced following the introduction of new health interventions in some settings. For instance, a polio vaccination program was rejected in a community in northern Nigeria due to wrong perception of religious leaders (*Jegede, 2007*). A similar experience was recorded in Ghana where community members rejected a mass deworming program scheduled by the government (*Dodoo et al., 2007*). In both instances, misunderstanding of the programs was responsible for their unsuccessful implementation (*Febir et al., 2013*). It is therefore evident that perception shapes one's knowledge of an infection and the acceptance of vaccination for its prevention. The Health

Belief Model also posits that high levels of perceived susceptibility to an infection increases the likelihood for adopting and accepting disease-preventive measures (*Tarkang & Zotor, 2015*). This array of evidence therefore indicates the need for evaluating the perception and practices of individuals prior to the introduction of a health intervention for each illness.

The uptake of vaccines and treatment options for illnesses have been described as an outplay of the cost evaluation in such regard among community members (*Hajizadeh, 2018*). Direct costs defined as the exact cost borne for the procurement of vaccines could be borne by a third party for example, the government to improve the uptake of vaccines (*Chukwuocha et al., 2018*); however, the uptake of vaccines may remain yet unsatisfactory. Unsatisfactory levels of vaccine uptake could result from the indirect costs attached to receiving such vaccines. Indirect costs such as transportation expenses to the health facility, loss of productive hours during vaccination waiting time, and registration bills at health facilities could deter the acceptance of vaccination programs to reasonable levels (*Hajizadeh, 2018*). In the COVID-19 context, indirect costs could limit the prospective vaccine uptake despite the direct costs being borne by a sponsoring body. Although associated costs cannot be completely borne, the COVID-19 vaccine sponsoring body would need to ensure the decentralization of vaccine collection points to existing primary health centers available in community settings.

Given the novelty of COVID-19, its associated fatality, and ongoing efforts for the development of an effective COVID-19 vaccine, it therefore becomes needful to examine the knowledge, attitudes, and practices of community members in this regard. Findings from this study would be helpful for adequate planning for the introduction of the COVID-19 vaccine. This formative study would thus be important in quickening prompt interventions which would be targeted at stimulating the right kind of support at community levels. This study therefore aimed to assess the willingness to pay and perception of community members in Oyo State, Nigeria regarding the COVID-19 vaccine.

## MATERIALS AND METHODS

### Study design and study setting

We conducted a descriptive cross-sectional study. Data were collected using an interviewer-administered questionnaire. Scheduled data collection took place between the 21st and 25th of September, 2020. We conducted the study in Ibadan, Oyo State, Nigeria. Ibadan is the third most populated city, and the largest city by geographical area in Nigeria. Ibadan is located 128 km inland northeast of Lagos and 530 km southwest of Abuja, the Federal Capital Territory. As of 31st December, 2020, Oyo State ranked fifth on the states affected by COVID-19 with 3,939 COVID-19 cases and 46 deaths recorded on the NCDC COVID-19 reports (*NCDC, 2020*). The *lingua franca* in Nigeria is English Language, and the major local language frequently used for communication in Ibadan is Yoruba.

## Study population

All community members aged 15 years and above from the selected communities in Ibadan were eligible to participate in the study. All individuals who provided verbal informed consent were included in the study. Community members less than 15 years were excluded because parental consent which would be required may not be possible due to parental absence when data collection was ongoing. We obtained verbal consent from all study participants.

## Sample size determination and sampling technique

We calculated the sample size using the formula for descriptive cross-sectional studies. The sample size was determined by the Leshlie Kish formula for sample determination for a single proportion as shown below:

$n = Z_\alpha^2 * p (1 - p)/d^2$ where:

$n$ = Minimum desired sample size

$Z_\alpha$ = the standard normal deviate, usually set as 1.96 which corresponds to a 5% level of significance.

$p$ = 50% was used

$d$ = Degree of accuracy (precision) set at 5% (0.05)

We adjusted for a 10% non-response rate, and therefore generated a total sample size of 440 respondents.

We selected study respondents using a multi-stage sampling technique. In stage 1, simple random sampling was used to select 4 out of the 5 urban local government areas in Ibadan. The choice of urban LGAs was preempted by the knowledge that many COVID-19 hotspots in Ibadan are located in the urban LGAs. In stage 2, we selected a political ward from each of the selected LGAs. From each of the selected wards, we randomly chose a central location. The direction of movement of the interviewers was determined by spinning a bottle. From areas corresponding to the direction of the bottle top, all eligible adults who gave verbal informed consent were included in the study until 110 persons were interviewed in each LGA. Therefore, we sampled a total of 440 individuals across the communities in the selected wards.

## Data collection methods

The questionnaire had six sections.

Section A, named "Sociodemographic characteristics" included respondents' information such as age of respondents, sex, occupation, religion, highest level of education, ethnicity, marital status, average monthly income, and wealth quintile. The second section named "Knowledge of COVID-19" was used to elicit information on the knowledge of COVID-19 among community members. The third section, named "Knowledge of the prospective COVID-19 vaccine" provided details on the knowledge of community members regarding the prospective COVID-19 vaccine. Close-ended questions were asked on the knowledge of COVID-19 as well as the awareness of the prospective COVID-19 vaccine.

The fourth section, named "Perceptions about the prospective COVID-19 vaccine" elicited information on the perceptions of community members regarding the prospective COVID-19 vaccine. Eight questions were asked on the perception about COVID-19 vaccine using a five-point Likert scale ranging from "Strongly Agree" to "Strongly Disagree". The questions asked were as follows: "COVID-19 is a major public health problem requiring vaccine", "COVID-19 vaccine will prevent COVID-19", "COVID-19 vaccine should get administered to everyone", "COVID-19 vaccine is against our cultural belief", "COVID-19 vaccine will save productive hours lost to COVID-19 illness", "COVID-19 vaccine will save money spent on COVID-19 treatment", "I will take the COVID-19 vaccine when produced", and the "COVID-19 vaccine will not have adverse health effect". The fifth section, named "Willingness to pay for the prospective COVID-19 vaccine" examined the willingness of community members regarding payment for the prospective COVID-19 vaccine. Close-ended questions were asked on the willingness to pay for the COVID-19 vaccine and the intent to comply with the prospective COVID-19 vaccine. The questions included "Are you willing to pay for the COVID-19 vaccine?", "If yes, specify reasons for your willingness", "If no, specify reasons for your unwillingness", and "What maximum amount are you willing to pay for the vaccine?". The interviewer correctly noted the all points stated by the respondents. The sixth section, "Information required before accepting the prospective COVID-19 vaccine" provided details on the information community members required before willingness to accept the COVID-19 vaccine could be gained.

We adapted the questionnaire from a tool used in a similar perception study on malarial vaccine in Southeast Nigeria (*Chukwuocha et al., 2018*). Tool validation was done by an infectious disease epidemiologist. The questionnaire was pre-tested by the administration of 5 questionnaires in communities that were not selected for this study. We rephrased a few ambiguous questions. We back-translated the questionnaire using the competencies of experts who had an excellent grasp of the Yoruba language. We administered the questionnaire to most of the respondents in English language because a larger proportion of the study respondents had at least basic formal education. A postgraduate student was trained for data collection, and this helped to eliminate potential bias associated with administration of questionnaire by more individuals.

Independent variables included: Sociodemographic characteristics such as age, sex, level of education, occupation, and ethnic group.

Outcome/dependent variables were the knowledge of the prospective COVID-19 vaccine, perception regarding the prospective COVID-19 vaccine, the willingness to pay for the vaccine, and information required before accepting the prospective COVID-19 vaccine.

## Data management

The Statistical Program for Social Sciences (SPSS version 20) software was used to analyze the data after data entry and cleaning (IBM *Corp, 2011*). Age was summarized using mean and standard deviation, while frequencies and percentages were used for categorical variables. We assigned points of "1" and "0" to each correct and incorrect identified cause

of COVID-19 respectively for 5 questions on the causes of COVID-19. Using the Bloom's cut-off, individuals with 3 or more cumulative points were categorized to have good knowledge of the cause of COVID-19, while people with lower points therefore had poor knowledge of COVID-19 cause.

We calculated the wealth index of respondents using the Principal Component Analysis (PCA) in SPSS (IBM *Corp, 2011*). The input to the PCA included responses on ownership of house and other key assets such as a stove, electric fan, refrigerator, air conditioner, radio, television, and generator, piped water in the household, bicycle, motor vehicle, upholstered chairs, sewing machine and washing machine. Thereafter, we calculated distribution cut-off points using quintiles. The quintiles were Q1= first, Q2 = second, Q3 = third, Q4 = fourth, Q5 = fifth; with the poorest in the first wealth quintile and the richest in the fifth wealth quintile.

Individuals who have heard of the prospective COVID-19 vaccine were assigned a score of "1", while those who have not heard were assigned a score of "0". Among the respondents who have heard of the prospective vaccine, the sources of COVID-19 vaccine information were assessed. The perceptions of respondents about COVID-19 were assessed using the five-point Likert scale with options ranging from "Strongly Agree" to "Strongly disagree". We assigned a score of "1" to the "Strongly Agree" option, "2" to the "Agree" option; "3" to the "Not decided" option, "4" to the "Disagree" option, and "5" to the "Strongly disagree" option. At the point of data analysis, recoding of the five-point Likert scale was done for questions which had been asked in the positive direction. Therefore, we computed a score of "5" for the "Strongly Agree" option, "4" for "Agree", "3" for "Not decided", "2" for "Disagree", and "1" for the "Strongly disagree" option. Eight questions were asked on the perception of community members regarding the prospective COVID-19 vaccine for which a total of "40" points were obtainable. Using the Bloom's cut-off point, scores ≥32 points (≥80%) implied a positive perception, while those corresponding to <32 points (<80%) implied a negative perception regarding the prospective COVID-19 vaccine.

Chi-square test was used for the assessment of associations between sociodemographic characteristics and the willingness to pay for the prospective COVID-19 vaccine. Multivariate analysis of the determinants of willingness to pay for the prospective COVID-19 vaccine was conducted using the Logistic regression model. Since no data was collected at stages 1 and 2 of the sampling process, we built logistic regression without adjusting for clustering. *P*-values <0.05 were statistically significant.

## Ethical approval and consent to participate

We obtained ethical approval for this study as part of COVID-19 Knowledge, attitude, practice and perception studies from the Oyo State Ministry of Health Ethical Review Committee with reference number AD/13/479/1779A. Informed consent and/or assent where required was obtained from the respondents. All respondents were assured of the confidentiality of information obtained from them. The respondents were duly informed of their right to withdraw from the study prior to its completion without any adverse

implication. No known harm was inflicted on the respondents as a result of participation in this study.

## RESULTS

The mean age of the 440 respondents was 37.22 ± 15.36 years. Overall, 202 (45.90%) were aged between 21 and 40 years. Among the respondents, 193 (43.90%) were males, 293 (66.60%) practiced Christianity, 371 (84.30%) were Yoruba, and 285 (64.80%) were married. Other sociodemographic information is as shown in Table 1.

Among the respondents, 311 (70.70%) had good knowledge on the cause of COVID-19. The causes of COVID-19 stated included contacts with saliva from a COVID-19-infected person among 367 (83.40%) repondents and participating in burial rites of a person who has died from COVID-19 among 423 (96.10%). Other causes mentioned by respondents included contact with beddings, clothing, and personal utensils of a person who is sick of COVID-19, and respiratory droplets of an infected person. Also, 292 (67.30%) of the respondents had heard of the prospective COVID-19 vaccine. Among them, 205 (70.20%) had obtained the prospective COVID-19 vaccine information from the radio, while 201 (68.80%) had been informed on the prospective COVID-19 vaccine via the television. Also, 175 (59.90%) respondents were informed of the COVID-19 vaccine through the social media. Other sources of information on the prospective COVID-19 vaccine are as shown in Fig. 1.

Table 2 shows the perceptions on the prospective COVID-19 vaccine among respondents. Among the respondents, 281 (96.20%) strongly agreed that COVID-19 is a major public health problem requiring vaccine, while 279 (95.50%) strongly agreed that the COVID19 vaccine would prevent COVID-19. Also, 182 (62.30%) strongly disagreed that the COVID-19 vaccine is against their cultural belief, and 180 (61.60%) strongly agreed to take the COVID-19 vaccine when produced.

Overall, 232 (79.50%) respondents had a positive perception regarding COVID-19 vaccine compared to 60 (20.50%) with negative perception. Eighty-one (18.40%) of the respondents were willing to pay for the prospective COVID-19 vaccine, among whom 45 (55.6%) were willing to pay at least 5000 naira ($13.16). All 81 (100.00%) respondents who were willing to pay for the COVID-19 vaccine attributed their willingness on the need to stay healthy. All 359 (100.00%) respondents who were unwilling to pay for the vaccine attributed their unwillingness to the unaffordability of vaccine costs by households. Also, 275 (62.50%) respondents required specific information on the prospective COVID-19 vaccine before accepting it (Table 3).

Fourteen (15.90%) respondents who belonged to the fourth wealth quintile were willing to pay for the prospective COVID-19 vaccine compared to 74 (84.10%) within same wealth quintile who were unwilling to pay. Forty-eight (54.50%) respondents in the fifth wealth quintile were willing to pay for the prospective COVID-19 vaccine compared to 40(45.50%) who were unwilling to pay ($X^2$ = 99.32, $p$ = <0.01). Individuals in the fourth wealth quintile were twice more likely to be willing to pay for the COVID-19 vaccine compared to those in the first wealth quintile (Adjusted Odds Ratio = 2.22, 95% CI [0.66–7.44], $p$ = 0.20). Individuals in the fifth wealth quintile were ten times more

**Table 1 Socio-demographic characteristics of community members in Ibadan, Nigeria.**

| Socio-demographic characteristics | Frequency | % | 95% Confidence interval | |
|---|---|---|---|---|
| | | | Upper | Lower |
| Age group (Years) | | | | |
| ≤20 | 68 | 15.50 | 12.40 | 19.10 |
| 21–40 | 202 | 45.90 | 41.30 | 50.60 |
| 41–60 | 131 | 29.80 | 25.70 | 34.20 |
| >60 | 39 | 8.90 | 6.60 | 11.90 |
| Sex | | | | |
| Male | 193 | 43.90 | 39.30 | 48.50 |
| Female | 247 | 56.10 | 51.50 | 60.70 |
| Religion | | | | |
| Christianity | 293 | 66.60 | 61.40 | 70.20 |
| Islam | 145 | 33.00 | 28.70 | 37.50 |
| Traditional | 2 | 0.50 | 0.12 | 1.64 |
| Highest level of education | | | | |
| Primary and below | 64 | 14.50 | 11.60 | 18.10 |
| Secondary and above | 376 | 85.50 | 81.90 | 88.40 |
| Ethnicity | | | | |
| Yoruba | 371 | 84.30 | 80.60 | 87.40 |
| Ibo | 59 | 13.40 | 10.50 | 16.90 |
| Hausa | 10 | 2.30 | 1.20 | 4.10 |
| Occupation | | | | |
| Business/trader | 162 | 36.80 | 32.40 | 41.40 |
| Artisan | 101 | 23.00 | 19.27 | 27.11 |
| Professional/civil servant/teacher | 68 | 15.50 | 12.38 | 19.13 |
| Retiree/housewife/cleric/student | 109 | 24.80 | 20.97 | 29.01 |
| Marital status | | | | |
| Married | 285 | 64.80 | 60.20 | 69.10 |
| Single | 132 | 30.00 | 25.91 | 34.44 |
| Others* | 23 | 5.20 | 3.51 | 7.72 |
| Average monthly income | | | | |
| <30,000 naira | 149 | 33.90 | 29.60 | 38.41 |
| ≥30,000 naira | 291 | 66.10 | 61.59 | 70.40 |
| Wealth quintiles | | | | |
| First | 88 | 20.00 | 16.53 | 23.99 |
| Second | 88 | 20.00 | 16.53 | 23.99 |
| Third | 88 | 20.00 | 16.53 | 23.99 |
| Fourth | 88 | 20.00 | 16.53 | 23.99 |
| Fifth | 88 | 20.00 | 16.53 | 23.99 |

**Note:**
  * Widowed/divorced.

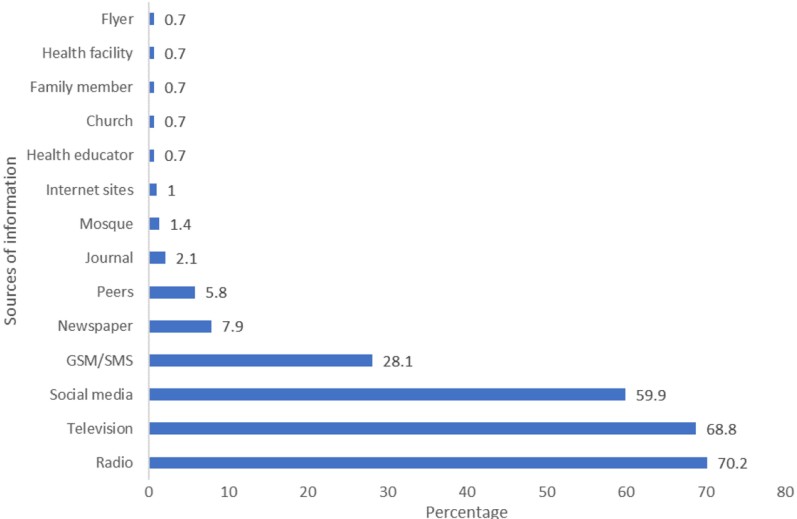

**Figure 1  Sources of information on the prospective COVID-19 vaccine among community members in Ibadan, Nigeria.**

likely to be willing to pay for the prospective COVID-19 vaccine compared to those in the first wealth quintile (Adjusted Odds Ratio = 9.57, 95% CI [2.88–31.82], $p$ = <0.01). Other determinants of the willingness to pay for the COVID-19 vaccine are as shown in Table 4.

## DISCUSSION

This study found that a large proportion of individuals (100.00%) were aware of the COVID-19 infection. Such a level of awareness is expected because COVID-19 occurrence is not a completely new event in Nigeria. Nigeria has been faced with the COVID-19 pandemic since the 27th of February, 2020, and implemented some mitigation measures regarding the containment of the COVID-19 infection. In this study, we found that many individuals (67.30%) are aware of the prospective COVID-19 vaccine. This finding could be possibly explained by the higher proportion of individuals with secondary education and above enrolled in this study. Some literatures have also reported the positive relationship between education and health awareness (*Sani, Naab & Aziato, 2016*; *Wang et al., 2018*). Education may therefore be an important predictor of the awareness of prospective health interventions in communities with more educated persons. However, alternate channels of information could be employed in communicating intended health interventions across all educational levels in communities.

Regarding the source of information on the prospective COVID-19 vaccine, traditional media such as the radio and television provided more information to more individuals compared to other channels of information dissemination. Other studies have reported the dominance of traditional media in communicating COVID-19-related information (*Olapegba et al., 2020*; *Ilesanmi & Afolabi, 2020a*). The social media, a modern channel of information source, also accounted for nearly two-thirds of COVID-19 vaccine information. Findings from Egypt however reported that Facebook, a modern information site mainly provided information on COVID-19 to her citizens (*Abdelhafiz et al., 2020*).

**Table 2 Perceptions on the prospective COVID-19 vaccine among community members in Ibadan, Nigeria.**

| Perception | Frequency | % | 95% Confidence interval | |
| --- | --- | --- | --- | --- |
| | | | Upper | Lower |
| COVID-19 is a major public health problem requiring vaccine (Total = 292) | | | | |
| Strongly agreed | 281 | 96.20 | 93.38 | 97.88 |
| Agreed | 2 | 0.70 | 0.19 | 2.46 |
| Not decided | 3 | 1.00 | 0.35 | 2.98 |
| Disagree | – | – | – | – |
| Strongly disagreed | 6 | 2.10 | 0.95 | 4.41 |
| COVID-19 vaccine will prevent COVID-19 (Total = 292) | | | | |
| Strongly agreed | 279 | 95.50 | 92.53 | 97.38 |
| Agreed | 3 | 1.00 | 0.35 | 2.98 |
| Not decided | 4 | 1.40 | 0.53 | 3.47 |
| Disagree | – | – | – | – |
| Strongly disagreed | 6 | 2.10 | 0.95 | 4.41 |
| COVID-19 vaccine should get administered to everyone (Total = 292) | | | | |
| Strongly agreed | 209 | 71.60 | 66.15 | 76.44 |
| Agreed | – | – | – | – |
| Not decide | 11 | 3.80 | 2.12 | 6.62 |
| Disagree | 26 | 8.90 | 6.15 | 12.73 |
| Strongly disagreed | 46 | 15.80 | 12.02 | 20.37 |
| COVID-19 vaccine is against our cultural belief (Total = 238) | | | | |
| Strongly agreed | 31 | 10.60 | 9.33 | 17.90 |
| Agreed | 35 | 12.00 | 10.77 | 19.77 |
| Not decide | 44 | 15.10 | 14.07 | 23.91 |
| Disagree | – | – | – | – |
| Strongly disagreed | 182 | 62.30 | 70.69 | 81.41 |
| COVID-19 vaccine will save productive hours lost to COVID-19 illness (Total = 292) | | | | |
| Strongly agreed | 270 | 92.50 | 88.86 | 94.97 |
| Agreed | 4 | 1.40 | 0.53 | 3.47 |
| Not decided | 13 | 4.50 | 2.62 | 7.47 |
| Disagree | – | – | – | – |
| Strongly disagreed | 5 | 1.70 | 0.73 | 3.95 |
| COVID-19 vaccine will save money spent on COVID-19 treatment (Total = 292) | | | | |
| Strongly agreed | 272 | 93.20 | 89.66 | 95.52 |
| Agreed | 3 | 1.00 | 0.35 | 2.98 |
| Not decided | 12 | 4.10 | 2.37 | 7.04 |
| Disagree | – | – | – | – |
| Strongly disagreed | 5 | 1.70 | 0.73 | 3.95 |
| I will take the vaccine when produced (Total = 292) | | | | |
| Strongly agreed | 180 | 61.60 | 55.95 | 67.04 |
| Agreed | 4 | 1.40 | 0.53 | 3.47 |

| Perception | Frequency | % | 95% Confidence interval | |
|---|---|---|---|---|
| | | | Upper | Lower |
| Not decided | 76 | 26.00 | 21.33 | 31.35 |
| Disagree | 9 | 3.10 | 1.63 | 5.75 |
| Strongly disagreed | 23 | 7.90 | 5.31 | 11.54 |
| COVID-19 vaccine will not have adverse health effects (Total = 292) | | | | |
| Strongly agreed | 133 | 45.50 | 39.93 | 51.28 |
| Agreed | 3 | 1.00 | 0.35 | 2.98 |
| Not decided | 147 | 50.30 | 44.64 | 56.04 |
| Disagree | 2 | 0.70 | 0.19 | 2.46 |
| Strongly disagreed | 7 | 2.40 | 1.17 | 4.86 |

In addition, the internet, a social media platform, provided more Undergraduate students in Jordan with information on COVID-19 (*Olaimat et al., 2020*). This finding therefore highlights the need for harnessing these channels of information dissemination with high coverage to communicate rich information on the COVID-19 vaccine. Due to the aforementioned reasons, the Nigeria Center for Disease Control utilizes both the traditional and social media platforms for communicating COVID-19 information (*Adepoju, 2020*; *Sote, 2020*). In the COVID-19 vaccine context, it is required that collaboration be implemented across these platforms for the timely dissemination of information to members of the public. Health facilities should also be equipped with up-to-date information on the prospective COVID-19 vaccine for dissemination to individuals on hospital visits.

We found that many individuals acknowledged that COVID-19 is a public health problem requiring vaccine, and were confident that the COVID-19 vaccine will prevent COVID-19. The demonstration of such levels of assurance could be described as an outplay of the positive results gained from previous vaccination programs such as oral polio vaccination (OPV), measles, and yellow fever (*Doherty et al., 2016*). These vaccination programs led to a drastic reduction in the incidence of these illness, and helped to maintain healthy conditions in children (*Febir et al., 2013*; *Chukwuocha et al., 2018*). Many respondents strongly agreed that the COVID-19 vaccine will save productive hours and money lost to the COVID-19 illness. Loss of productive hours in the COVID-19 context has been attributed to include the turn-around time for collection of COVID-19 test results, and time spent on isolation (*Ilesanmi & Afolabi, 2020b*, *2020c*). In spite of these potential benefits presented by the prospective COVID-19 vaccine, fewer persons however expressed their willingness to take the COVID-19 vaccine. Such unwillingness for vaccine acceptance stemmed from the skepticism associated with the affordability of the COVID-19 vaccine by households if costs were involved.

The minimum monthly wage of 30, 000 naira ($78.95) received by many Nigerians is an evidence that if COVID-19 vaccine costs exceeds 5,000 naira ($13.16), such

Table 3 **Willingness to pay for the COVID-19 vaccine and COVID-19 information required by community members in Ibadan, Nigeria.**

| | n | % | 95% Confidence interval | |
| --- | --- | --- | --- | --- |
| | | | Upper | Lower |
| Willingness to pay for the COVID-19 vaccine | | | | |
| Yes | 81 | 18.40 | 15.07 | 22.30 |
| No | 359 | 81.60 | 77.70 | 84.93 |
| Maximum amount intended for payment (Total = 81) | | | | |
| <5,000 naira ($13.16) | 36 | 44.40 | 34.12 | 55.27 |
| ≥5,000 naira ($13.16) | 45 | 55.60 | 44.73 | 65.88 |
| Reasons for willingness* | | | | |
| To stay healthy | 81 | 100.00 | 95.47 | 100.00 |
| To prevent loss of productive hours | 23 | 28.40 | 19.73 | 39.02 |
| To prevent further treatment expenses | 23 | 28.40 | 19.73 | 39.02 |
| To promote social acceptability of vaccines | 9 | 11.10 | 5.96 | 19.79 |
| Reasons for unwillingness* | | | | |
| Costs not affordable by households | 359 | 100.00 | 98.98 | 100.00 |
| Fear of adverse effects | 30 | 8.40 | 5.92 | 11.68 |
| Fear of inaccessibility of vaccines | 2 | 0.60 | 0.04 | 2.38 |
| Contrary to religious beliefs | 16 | 4.50 | 2.76 | 7.12 |
| Contrary to culture | 1 | 0.30 | 0.01 | 1.31 |
| Require specific information on COVID-19 vaccine | | | | |
| Yes | 275 | 62.50 | 57.89 | 66.90 |
| No | 165 | 37.50 | 33.10 | 42.11 |
| Information required before accepting COVID-19 vaccine* | | | | |
| Whether payments would be required | 248 | 90.20 | 86.09 | 93.16 |
| Possible side effects of the vaccine | 175 | 63.60 | 57.80 | 69.10 |
| Number of doses needed | 131 | 47.60 | 41.81 | 53.53 |
| Whether the vaccine will prevent or cure COVID-19 | 90 | 32.70 | 27.45 | 38.48 |
| Route of administration | 58 | 21.10 | 16.68 | 26.29 |
| Age range of individuals to be vaccinated | 53 | 19.30 | 15.05 | 24.35 |
| Manufacturer of the vaccine | 24 | 8.70 | 5.93 | 12.66 |
| Vaccine collection points | 17 | 6.20 | 3.90 | 9.68 |
| Duration of immunity provided | 11 | 2.50 | 2.25 | 7.02 |
| Whether vaccination would be accompanied by incentives | 7 | 2.70 | 1.24 | 5.16 |
| Vaccine's expiry date | 2 | 0.50 | 0.20 | 2.61 |

**Note:**
* Multiple responses allowed.

procurement may not be affordable to the average Nigerian. Non-compliance to health interventions especially in low-resourced settings have been linked to the costs and affordability of such interventions. This has therefore limited the successes achieved on priority illnesses, such as malaria (*Chukwuocha et al., 2018*). Health interventions with no attached healthcare costs have achieved better results (*Chukwuocha et al., 2018*).

**Table 4 Association between sociodemographic characteristics and willingness to pay for the prospective COVID-19 vaccine among community members in Ibadan, Nigeria.**

| Socio-demographic characteristics | Willingness to pay | | Adjusted odds ratio (95% confidence interval) | p-value |
|---|---|---|---|---|
| | Yes frequency (%) | No frequency (%) | | |
| **Age group (Years)** | | | | |
| ≤20 | 0 (0.00%) | 60 (100.00) | 0.0 [<0.01–<0.01] | 1.00 |
| 20–39 | 27 (13.20) | 177 (86.80) | 0.82 [0.40–1.71] | 0.60 |
| ≥40 | 54 (30.70) | 122 (69.30) | 1 | |
| | $X^2 = 34.82$ | $p = <0.01$ | | |
| **Sex** | | | | |
| Male | 40 (20.70) | 153 (79.30) | | |
| Female | 41 (16.60) | 206 (83.40) | | |
| | $X^2 = 1.23$ | $p = 0.27$ | | |
| **Highest level of education** | | | | |
| Primary and below | 5 (7.80) | 59 (92.20) | 0.50 [0.16–1.53] | 0.23 |
| Secondary and above | 76 (20.20) | 300 (79.80) | 1 | |
| | $X^2 = 5.50$ | $p = 0.02$ | | |
| **Ethnicity** | | | | |
| Yoruba | 66 (17.80) | 305 (82.20) | | |
| Ibo | 15 (25.40) | 44 (74.60) | | |
| Hausa | 0 (0) | 10 (100.00) | | |
| | $X^2 = 4.28$ | $p = 0.12$ | | |
| **Occupation** | | | | |
| Business/trader | 32 (19.80) | 130 (80.20) | 0.61 [0.27–1.39] | 0.24 |
| Artisan | 22 (21.80) | 79 (78.20) | 0.50 [0.23–1.07] | 0.08 |
| Professional/civil servant/teacher | 36 (38.20) | 42 (61.80) | 0.05 [0.01–0.52] | 0.01 |
| Retiree/housewife/cleric/student | 1 (0.90) | 108 (99.10) | 1 | |
| | $X^2 = 40.96$ | $p = <0.01$ | | |
| **Marital status** | | | | |
| Married | 72 (25.30) | 213 (74.70) | 1.17 [0.19–7.08] | 0.87 |
| Single | 5 (3.80) | 127 (96.20) | 1.35 [0.43–4.28] | 0.61 |
| Others* | 4 (17.40) | 19 (82.60) | 1 | |
| | $X^2 = 27.72$ | $p = <0.01$ | | |
| **Average monthly income** | | | | |
| <30,000 | 5 (3.40) | 144 (96.60) | 1.45 [0.45–4.66] | 0.53 |
| ≥30,000 | 76 (26.10) | 215 (73.90) | 1 | |
| | $X^2 = 33.99$ | $p = <0.01$ | | |
| **Wealth quintiles** | | | | |
| First | 5 (5.70) | 83 (94.30) | 1 | |
| Second | 6 (6.80) | 82 (93.20) | 1.21 [0.32–4.60] | 0.79 |
| Third | 8 (9.10) | 80 (90.90) | 1.14 [0.32–4.12] | 0.84 |
| Fourth | 14 (15.90) | 74 (84.10) | 2.22 [0.66–7.44] | 0.20 |
| Fifth | 48 (54.50) | 40 (45.50) | 9.57 [2.88–31.82] | <0.01 |
| | $X^2 = 99.32$ | $p = <0.01$ | | |

**Note:**
* Divorced/Widowed, $p < 0.01$.

Affordability by households should therefore be one of the factors given precedence during the planning and implementation of the prospective COVID-19 vaccine production. In addition, consideration should be given to all income groups in the population so that no population subgroup would be excluded from assessing the prospective COVID-19 vaccine program.

Among the respondents who would require specific information on the prospective COVID-19 vaccine, information on payments was the most frequently stated required information. This posits that the costs attached could either reduce or increase the uptake of the COVD-19 vaccine when produced. Many individuals would also require information on the possible side effects before accepting the COVID-19 vaccine. Although it is known that many existing vaccines have minimal levels of side effects such as temporary diarrhea (*CDC, 2018*), the novelty of the prospective COVID-19 vaccine necessitates specific information on its side effects. If the possible side effects of the prospective COVID-19 vaccine are not too different from the side effects experienced with other illnesses for which vaccines are received, more individuals are likely to accept the prospective COVID-19 vaccine. Studies conducted on malarial vaccine have similarly documented side effects as an inevitable factor which influences the acceptance and compliance with the malarial vaccine (*Menaca et al., 2014*; *Abdulkadir & Ajayi, 2015*). The side effects of the prospective COVID-19 vaccine (if any) should be communicated alongside COVID-19 mitigation measures on the radio, TV, internet sites, and health facilities to ensure that no one is excluded regarding the COVID-19 vaccine information.

This study found that occupation is an important determinant to the willingness to pay for the COVID-19 vaccine. We similarly found that wealth index also determines the willingness to pay for the COVID-19 vaccine. This finding therefore implies that individuals in the higher wealth quintile are willing to pay for the prospective COVID-19 vaccine primarily because they can afford it. Building on the foregoing, persons in the lower wealth quintile would be missed out on in the implementation of the prospective COVID-19 vaccine if only the higher wealth quintiles are considered regarding affordability of the COVID-19 vaccine. Previous studies conducted on malarial vaccine did not report any association between occupation or wealth index and willingness to pay for the vaccine (*Menaca et al., 2014*; *Abdulkadir & Ajayi, 2015*). In view of the present study, the COVID-19 pandemic has greatly affected the income of many individuals, and this could be an explanation for this finding. This finding further posits the need for the subsidization of the COVID-19 vaccine to improve the uptake of the vaccine when available.

## Strengths of the study

Up-to-date, majority of COVID-19 researches have been conducted on the knowledge, attitude, and practices of population groups on the COVID-19 illness itself. In line with recent developments on the containment and prevention of the COVID-19 infection, the present study has gone a step further in assessing the perception and willingness to

pay for the prospective COVID-19 vaccine. To the best of our knowledge, this is the first of its kind. We also ruled out bias associated with multiple data collectors or the use of electronic data collection tools by using only one interviewer for data collection.

### Limitations of the study

Firstly, the study respondents were largely literate. The findings from this study therefore may not be generalizable in a less-literate setting. Also, the use of a small sample size limited the results obtained during further analysis, resulting in an extremely large confidence interval.

## CONCLUSION

The perception of the prospective COVID-19 vaccine determines the willingness to take the COVID-19 vaccine. Individuals may be willing to take the COVID-19 vaccine, however the cost of purchasing it may not be affordable. It is therefore required that the prospective COVID-19 vaccine is fully subsidized or freely administered in order to encourage its uptake among all individuals. In addition, information on the prospective COVID-19 vaccine and possible adverse effects should be adequately communicated in clear terms through different channels of information such as TV and radio stations, social media, and health facilities. This will aid the implementation, acceptance, and compliance to the prospective COVID-19 vaccine, and will aid the sustainable journey towards the elimination of the COVID-19 pandemic. Further research should be conducted across COVID-19 affected countries to assess the preparedness of community members towards the eventual roll-out of the prospective COVID-19 vaccine.

## ACKNOWLEDGEMENTS

The authors are grateful to all the community members who participated in this research.

### Funding

The authors received no funding for this work.

### Competing Interests

The authors declare that they have no competing interests.

### Author Contributions

- Olayinka Ilesanmi conceived and designed the experiments, performed the experiments, analyzed the data, authored or reviewed drafts of the paper, and approved the final draft.
- Aanuoluwapo Afolabi conceived and designed the experiments, performed the experiments, analyzed the data, prepared figures and/or tables, authored or reviewed drafts of the paper, and approved the final draft.
- Obioma Uchendu analyzed the data, authored or reviewed drafts of the paper, and approved the final draft.

## Human Ethics

The following information was supplied relating to ethical approvals (i.e., approving body and any reference numbers):

The Oyo State Ministry of Health Ethical Review Committee granted ethical approval to carry out the study (AD/13/479/1779A).

## Data Availability

Raw data, including the sociodemographic characteristics and perception of Ibadan community members regarding the prospective COVID-19 vaccine, are available in the Supplemental Files.

## Supplemental Information

Supplemental information for this article can be found online at http://dx.doi.org/10.7717/peerj.11153#supplemental-information.

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
