# Peer review of "The prospective COVID-19 vaccine: willingness to pay and perception of community members in Ibadan, Nigeria"

_PeerJ, doi:10.7717/peerj.11153_

## Round 0.1 · original submission · Major Revisions

Thank you for submitting your research for consideration to PeerJ. Experts have reviewed your manuscript and provided excellent comments that should improve on your research reporting. Please pay attention to the comments regarding improving on the statistical methods section in terms of choice of statistical tests or models, clearly describing the analysis details and interpretation of findings.

·

Basic reporting

The manuscript is well written and easy to follow with appropriate and sufficient context provided. The use of English is appropriate.

Experimental design

Epidemiologically, appropriate Study and sampling design were used. All the expected procedures and technicalities of the use of questionnaires for an epidemiologic study were well observed.

Validity of the findings

The findings of the study are internally valid for the study population.

Additional comments

Please check throughout the manuscript for error in spellings especially COVOD 19 instead of COVID 19.
Line 165-198: You mentioned PCA was done but no result was shown. Also, I didnt see any inclusion of any principal component for socioeconomic index in the multivariable logistic regression model. OR do you mean line 169-170 represent this socioeconomic index? A more explicit explanation and coherent information in this section will avoid any ambiguity.
Please kindly dd 95% CI interval to Tables 1-3
Table 1-4: Please be consistent with your decimal places, You used 1,2 and 3 decimal places in all these tables.
Line 154-156: You indicated here that you have 2-3 outcomes of interest, why was only multivariable model done for only willingness to pay?
Line 117-127: Uf you look at your data and sampling design, you have 3 level hierarchical data structure with center location nested in political ward nested in the local government. On a good day, you should have accounted for this clustering in the data by building mixed effect logistic model, however considering no data was collected at stage 1 and stage 2, its permissible to build logistic regression without adjusting for clustering. Please add this explanation to the data analytic part of the manuscript.

·

Basic reporting

Figures are relevant, high quality, well labelled & described.
No, detailed description of the figure should be provided. Description should include percentage of respondents. The Y-axis was labeled as "Proportion" while the figure seems to display percentages. Please modify appropriately.
Raw data supplied.
Authors provided raw data, but no code

Experimental design

Rigorous investigation performed to a high technical & ethical standard.
The authors can do better by adding more details in the statistical analysis section.
Methods described with sufficient detail & information to replicate.
More detailed statistical analysis will be of benefit to reader.

Validity of the findings

No comment

Additional comments

General comments
Abstract
Lines 11-13: Please, clearly state what score was assigned to each of the 5-point Likert scale.
Line 16: Please state what kind of multivariate analysis was conducted.
Line 19: Insert the unit for mean age
Line 20: Delete “and”
Line 23: Define AOR
Introduction
Line 37-40: Insert reference for the statement “In spite of these containment and control efforts, ……”
Line 79: In line with the study aim which include assessing the willingness to pay for the COVID 19 vaccine, the introduction section should be expanded to address cost of vaccine and vaccination on persons in community and its impact on willingness of the people to take it due to economic reasons
Materials and methods
Line 93: Insert space - 128 kilometres ......... 530 kilometres
Lines 94-95: Insert reference. “As of 28th September, 2020, Oyo State…..”
Line 112: Delete “be”
Lines 116-127: Consider writing this section in full sentence format
Line 118: What informed the selection of 4 out of 11 local government areas?
Line 126-127: A total of 440 persons sampled across 4 LGAs. Does this correspond to 440 households? Please clearly state how many households is represented in the sampled respondents.
Lines 129-137: Consider writing this section in full sentence format
Line 139-140: Consider stating the eight questions here OR reference the questionnaire in the supplementary material.
Line 158: State the computer program used for data entry.
Lines 165-166: Insert reference(s).
Lines 180-182: Repeat here that there were 8 questions addressing the perceptions of respondents about COVID-19, summing up to a possible 40 points score.
Line 183: Consider “≥ 32 points (≥ 80%)” for consistency
Line 187: “COVID-19vaccine” insert space
Lines 223-225: It might be interesting to understand the demographic distributions of the 311 respondents who had good knowledge. Is it different for the respondents with poor knowledge?
Lines 225-227: These are causes of COVID-19 predetermined by the authors in the interviewer-administered questionnaire. Did the respondents mentioned other causes of COVID-19 besides the ones listed in the questionnaire? What are the other causes of COVID-19 mentioned by respondents?
Line 237: Clearly state who you described here as "them". Questionnaire respondents?
Line 244: Consider including the dollar equivalent of this naira amount.
Lines 250-252: Add “%” to the numbers in parentheses
Discussions
Line 283: “..two -thirds of COVID-19..” remove space
Line 314: Insert reference(s)
Lines 340-342: Insert reference(s)
Lines 344-345: Consider adding "when available" at the end of the sentence ".....improve the uptake of the vaccine"
Line 361: “COVID-19 vaccine”, not COVOD-19 vaccine.
References
Lines 402-404: Insert version and page number
Line 416: “affect” not affect
Tables
Define "n" in tables 2-4
Consider detailed description of the tables in the titles
Figure
Consider detailed description for the figure. Description should include percentage of respondents. The Y-axis was labeled as "Proportion" while the figure seems to display percentages. Please modify appropriately.

---

## Round 0.2 · Major Revisions

Experts have reviewed your revised manuscript and still found several issues that should be addressed.

·

Basic reporting

The manuscript is well written

Experimental design

The experimental design is sound and easy to follow.

Validity of the findings

Necessary corrections have been done by the authors with data and code provision.

Additional comments

Table 1: Sociodemographic characteristics, for the age group, the 95% CI (using exact method) is as follows:

<=20: 68/440 =15.5% (12.2-19.2)
21-40: 202/440 =45.9% (41.2-50.7)
41-60: 131/440 =29.8% (25.5-34.3)
>60: 39/440 =8.9% (6.3-11.9)

So adding CI is possible and it doesn't have to be a bivariate analysis. This is a binomial distribution with n and p and therefore, 95% CI can be calculated.

·

Basic reporting

No comment

Experimental design

No comment

Validity of the findings

No comment

Additional comments

Comments for minor revision included in the uploaded annotated PDF file.

---

## Round 0.3 · Major Revisions

Thank you for addressing the comments from reviewer 2. I may have missed it but I don't see the manuscript updated with the 95% CI for the estimates Tables 1, 2, 3 and in the proportions in Table 4, as reviewer #1 requested. The 95% CI as the measure of uncertainty for your estimates here is a standard reporting requirement and hence are important. Please update the manuscript accordingly and resubmit.

---

## Round 0.4 · Major Revisions

Dear authors,

Once again, I don't see the manuscript updated with the 95% CI for the estimates Tables 1, 2, 3 and in the proportions in Table 4.

The rebuttal document shows a table with answers to queries that seem to be from the editorial staff. Was the wrong version of the rebuttal uploaded and the previous manuscript version instead of the updated one?

To summarize, measures of uncertainty such as the 95% CI are a standard reporting requirement and without them readers cannot interpret your findings. Please update the manuscript accordingly and resubmit.

---

## Round 0.5 · accepted · Accept

Thank you for responding to my previous comments, I find your manuscript acceptable for publication.